# SHORT REPORT

# ERK inhibits Capicua repressor function via multisite phosphorylation

Sayantanee Paul[1,2,*], Khandan Ilkhani[1,*], Nathan Strozewski[1,*], Liu Yang[3], David W. Denberg[3,4,5], Wootchelmine Christalin[1], Vanessa Locke[1], Stanislav Y. Shvartsman[3,5,6] and Alexey Veraksa[1,‡]

## ABSTRACT

The receptor tyrosine kinase (RTK)/extracellular signal-regulated kinase (ERK) signaling pathway controls cell proliferation, differentiation and survival. The transcriptional repressor Capicua (Cic) has emerged as a key target for ERK-mediated downregulation in *Drosophila* and mammals, and pathogenic variants in human *CIC* result in cancer and neurological diseases. Phosphorylation by ERK (Rolled in flies) is critical for Cic downregulation, but the identities of phosphosites in *Drosophila* Cic are unknown. Here, we identify sites of phosphorylation in Cic that are directly targeted by ERK and validate their developmental functions *in vivo* using mutant Cic variants. Cic phosphosites are distributed throughout the length of the protein. Cic mutated in 20 high-confidence sites is resistant to proteasomal degradation and behaves as a 'super-repressor' *in vivo* that is largely insensitive to ERK-mediated downregulation. No single site is sufficient to turn off Cic activity; instead, we find that ERK must phosphorylate multiple sites in Cic simultaneously to achieve full downregulation. This multisite phosphorylation likely involves phosphodegrons that are recognized by ubiquitin ligases such as Ago (FBXW7 in mammals), contributing to Cic degradation. This study advances our understanding of the molecular mechanisms of signal interpretation downstream of the RTK/ERK signaling network.

KEY WORDS: Capicua, ERK, Multisite phosphorylation, Drosophila, Phosphodegron

## INTRODUCTION

Capicua (Cic) is a high-mobility group (HMG) box containing transcriptional repressor that acts downstream of the receptor tyrosine kinase (RTK)/extracellular signal-regulated kinase (ERK) signaling cascade [herein ERK refers to the fly protein Rolled (Rl), and mammalian ERK1 and ERK2, also known as MAPK3 and MAPK1, respectively]. In *Drosophila*, ERK-mediated Cic

[1]Department of Biology, University of Massachusetts Boston, Boston, MA 02125, USA. [2]Department of Discovery Oncology, Genentech Inc., South San Francisco, CA 94080, USA. [3]Lewis Sigler Institute for Integrative Genomics, Princeton University, Princeton, NJ 08544, USA. [4]Program in Quantitative and Computational Biology, Princeton University, Princeton, NJ 08544, USA. [5]Flatiron Institute, New York, NY 10010, USA. [6]Department of Molecular Biology, Princeton University, Princeton, NJ 08544, USA.
*These authors contributed equally to this work

‡Author for correspondence (alexey.veraksa@umb.edu)

 A.V., 0000-0003-2920-080X

phosphorylation and downregulation are necessary for proper patterning and growth of multiple tissues during development (Jimenez et al., 2012). In humans, pathogenic variants in *CIC* have been implicated in neurodegenerative disease spinocerebellar ataxia type 1 (SCA1) (Fryer et al., 2011; Lam et al., 2006), in the majority of oligodendroglioma cases and in other cancers (Kim et al., 2020; Okimoto et al., 2017; Simon-Carrasco et al., 2017; Tanaka et al., 2017). In both flies and mammals, Cic phosphorylation is a crucial regulatory event in RTK/ERK signal transduction. Previous studies have identified the C2 domain in *Drosophila* Cic as mediating its binding to ERK (Astigarraga et al., 2007), and this interaction is required for downregulation of its function as a transcriptional repressor. The C1 domain, located at the C-terminus, forms a novel DNA-binding interface together with the HMG box (Fores et al., 2017b; Webb et al., 2025). Several mechanisms have been proposed to explain Cic downregulation, including loss of DNA binding, export to cytoplasm, protein degradation and loss of binding to corepressors (Ajuria et al., 2011; Astigarraga et al., 2007; Bunda et al., 2019; Dissanayake et al., 2011; Grimm et al., 2012; Jimenez et al., 2012; Keenan et al., 2020; Lim et al., 2013; Okimoto et al., 2017; Rodriguez-Munoz et al., 2022; Tseng et al., 2007).

Given that all the proposed modes of Cic inactivation are dependent on post-translational events such as ERK-dependent phosphorylation, it is essential to determine the role of these phosphosites in the context of Cic downregulation. Previous studies have identified and validated several sites of phosphorylation in human CIC, but the majority are not directly targeted by ERK. Phosphorylation of S173 near the HMG box is carried out by the kinase p90RSK downstream of ERK activation and is necessary for establishing interactions with 14-3-3 proteins, which might interfere with DNA binding (Dissanayake et al., 2011) and increase the export of CIC from the nucleus to the cytosol (Ren et al., 2020). A homologous site in *Drosophila* Cic (S461) has been proposed to serve a similar function (Dissanayake et al., 2011). Additionally, phosphorylated S173 interacts with the ubiquitin ligase PJA1, leading to CIC degradation (Bunda et al., 2019). ERK-mediated phosphorylation of two other residues might prevent binding of a C-terminal nuclear localization signal to importin, which interferes with CIC nuclear localization (Dissanayake et al., 2011). CIC is also phosphorylated by Src on tyrosine residues, which promotes its nuclear export (Bunda et al., 2020). Recently, a few phosphosites in *Drosophila* Cic have been identified in the context of a global identification of ERK phosphorylation targets in the embryo (Yang et al., 2024).

Despite these data, the full complement of phosphosites in *Drosophila* Cic is unknown. Here, we carried out a mass spectrometry-based screen to identify the phosphosites in *Drosophila* Cic that are directly targeted by ERK. We have found that Cic is phosphorylated on multiple sites and validated their functional role *in vivo* using variants carrying combinations of

the mutated phosphosite residues. Mutation of the 20 high-confidence sites generated a 'super-repressor' variant that was essentially insensitive to ERK downregulation, whereas subsets of these sites gave only a partial resistance to ERK. This suggests that Cic is regulated by ERK via multisite phosphorylation, and many sites must be targeted for complete downregulation. Cic mutant variants were also resistant to proteasomal degradation *in vitro*. We propose that at least some of these ERK-dependent sites form phosphodegrons recognized by ubiquitin ligases, which might ultimately lead to Cic degradation via the proteasome.

## RESULTS AND DISCUSSION
### Identification of ERK-dependent sites of phosphorylation in Cic

To identify the sites in Cic that are directly phosphorylated by ERK, we performed an *in vitro* kinase reaction using full-length Cic protein tagged with streptavidin-binding peptide (SBP) (Yang et al., 2016). Cic–SBP was purified from stably transfected *Drosophila* S2 cells under the condition of MEK inhibition by PD0325901 (Ciuffreda et al., 2009), which was used to reduce any background phosphorylation by endogenous ERK (Fig. 1A). Cic–SBP was immobilized on streptavidin beads and subjected to a kinase reaction using bacterially purified activated dually phosphorylated rat ERK2 (dpERK) and ATP. As a control, Cic–SBP was incubated with dpERK in the absence of ATP. The control and experimental samples were washed, eluted and analyzed by mass spectrometry (Fig. 1A). This experiment was performed for a total of four biological replicates (Fig. 1B; Fig. S1; see Table S1).

### ERK-mediated multisite phosphorylation is required for Cic downregulation

The first round of phosphosite identification ('run1') yielded 15 high confidence ERK-dependent phosphosites (serine and threonine residues) that were present in the experimental sample but not in the control (Fig. 1C; Fig. S1). We used this set to dissect the contributions of individual sites to Cic downregulation. Fly lines were generated that carried *UAS-Cic-SBP* with substitutions of all 15 sites to non-phosphorylatable alanine residues [the $Cic^{(1-15)A}$ mutant] as well as constructs carrying subsets of these substitutions: five groups of three sites each [$Cic^{(1-3)A}$, $Cic^{(4-6)A}$, $Cic^{(7-9)A}$, $Cic^{(10-12)A}$ and $Cic^{(13-15)A}$] and a larger set containing the three central subsets [$Cic^{(4-12)A}$; Fig. 1C]. We then asked how the ERK signaling readout was affected *in vivo* by overexpressing these Cic phosphomutant variants in the developing wing. As EGFR signaling relieves Cic-mediated downregulation of downstream target genes and thus promotes wing vein development (Roch et al., 2002), we focused on the loss of venation pattern in the adult wing as an indicator of gain of function of Cic repressor activity. Overexpression of wild-type Cic using the wing pouch *MS1096-GAL4* driver (Capdevila and Guerrero, 1994) resulted in a moderate vein loss corresponding to a Cic gain-of-function effect (Fig. 1D,E,M), but expression of either the $Cic^{(1-15)A}$ or the $Cic^{(4-12)A}$ mutant led to a much more severe phenotype (Fig. 1F,G,M), suggesting that these two mutant proteins became resistant to downregulation by ERK. Among the triplet subsets, the $Cic^{(7-9)A}$ mutant had the strongest effect (Fig. 1H,M), followed by more C-terminally located $Cic^{(10-12)A}$ and $Cic^{(13-15)A}$ (Fig. 1I,J,M). Expression of $Cic^{(1-3)A}$ and $Cic^{(4-6)A}$ gave phenotypes that were not different from expression of wild-type Cic (Fig. 1K–M).

The overall effects of the various mutant site combinations are summarized in Fig. 1N. Although the $Cic^{(7-9)A}$ mutant caused the most severe vein loss out of the triplet combinations, it could not fully phenocopy the overall $Cic^{(1-15)A}$ or the $Cic^{(4-12)A}$ expression

phenotype. Therefore, it appears that multiple phosphorylation sites function together to mediate ERK-dependent downregulation of Cic repressor activity downstream of EGFR.

### Cic activity *in vivo* correlates with the degree of phosphosite substitution

Altogether, our *in vitro* kinase experiments followed by mass spectrometry identified 21 high-confidence phosphosites that were present in two or more of the four biological replicates (Fig. 1B; Figs S1, S2). Based on these data, we generated a $Cic^{20A}$ mutant variant that contained 13 sites previously included in $Cic^{(1-15)A}$ as well as seven sites that were not mutated in $Cic^{(1-15)A}$ (Fig. S1). Phosphorylation of T1059 was identified twice but was omitted from the mutagenesis because this site is located within the C2 ERK-binding domain. Mutation of this residue impaired the binding between dpERK and Cic (Astigarraga et al., 2007), so it was likely to generate a very strong mutant in combination with other sites and obscure the effects of the other phosphosite mutations. S461, which is targeted by S6kII, the *Drosophila* homolog of mammalian p90RSK (Dissanayake et al., 2011), was not phosphorylated in our data.

Based on the *in vivo* results using the triplet subsets from $Cic^{(1-15)A}$, the central region of Cic appeared to contain the sites most relevant for Cic downregulation (Fig. 1N). Phosphorylated serine and threonine residues are often encountered in phosphodegrons recognized by various E3 ubiquitin ligases (Holt, 2012; Welcker and Clurman, 2008). A previous study identified a motif in the middle region of Cic that conforms to the consensus recognition sequence (pTPPxpS/T) of the E3 ubiquitin ligase Archipelago (Ago), which is homologous to mammalian FBXW7 (Singh et al., 2022; Suisse et al., 2017). We identified three additional regions nearby that generally conform to this consensus sequence and made a corresponding construct (named $Cic^{6A}$) that specifically targeted the phosphosites included in these four putative phosphodegrons (Fig. 1O; Fig. S2).

We then sought to verify that the mutation of these phosphosites did not disrupt the binding between Cic and ERK, as the observed effects could be due simply to a loss of this interaction. To test this, we co-expressed V5-tagged wild-type and mutant Cic constructs with *Drosophila* ERK–Flag in S2 cells and assayed their binding by co-immunoprecipitation. As shown in Fig. 1P, mutations of the phosphosites included in these constructs did not affect the binding of Cic to ERK, suggesting that any phenotypes resulting from overexpression of these constructs would not stem from an inability of ERK to associate with these Cic variants.

Overexpression of wild-type Cic–mVenus in the wing using *MS1096-GAL4* resulted in a mild vein loss corresponding to Cic gain-of-function effect (Fig. 2A,B), but expression of either the $Cic^{6A}$ or $Cic^{20A}$ mutant led to a much more severe loss of veins, suggesting that these two mutant proteins are resistant to downregulation by ERK (Fig. 2C,D). Strikingly, $Cic^{20A}$ expression had a phenotype that was almost as strong as the one caused by the $Cic^{\Delta C2}$ mutant, which carries a deletion of the C2 domain and thus abrogates ERK interaction (Fig. 2D,E). To study Cic regulation in the wing at an earlier developmental stage, we analyzed the pattern of expression of the *CUASC-lacZ* reporter, which carries Cic-binding sites flanking the *UAS* cassette (Ajuria et al., 2011). When combined with the wing-specific *C5-GAL4* driver (Yeh et al., 1995), *CUASC-lacZ* expression recapitulates the pattern of proveins in the third-instar larval wing imaginal disc (Ajuria et al., 2011; Yang et al., 2016). Whereas expression of wild-type Cic–mVenus in this background still gave a complete overall pattern of proveins (Fig. 2F), expression of $Cic^{6A}$ led to a reduction in that pattern (Fig. 2G), and the expression of the $Cic^{20A}$ and $Cic^{\Delta C2}$ mutants very strongly inhibited LacZ expression

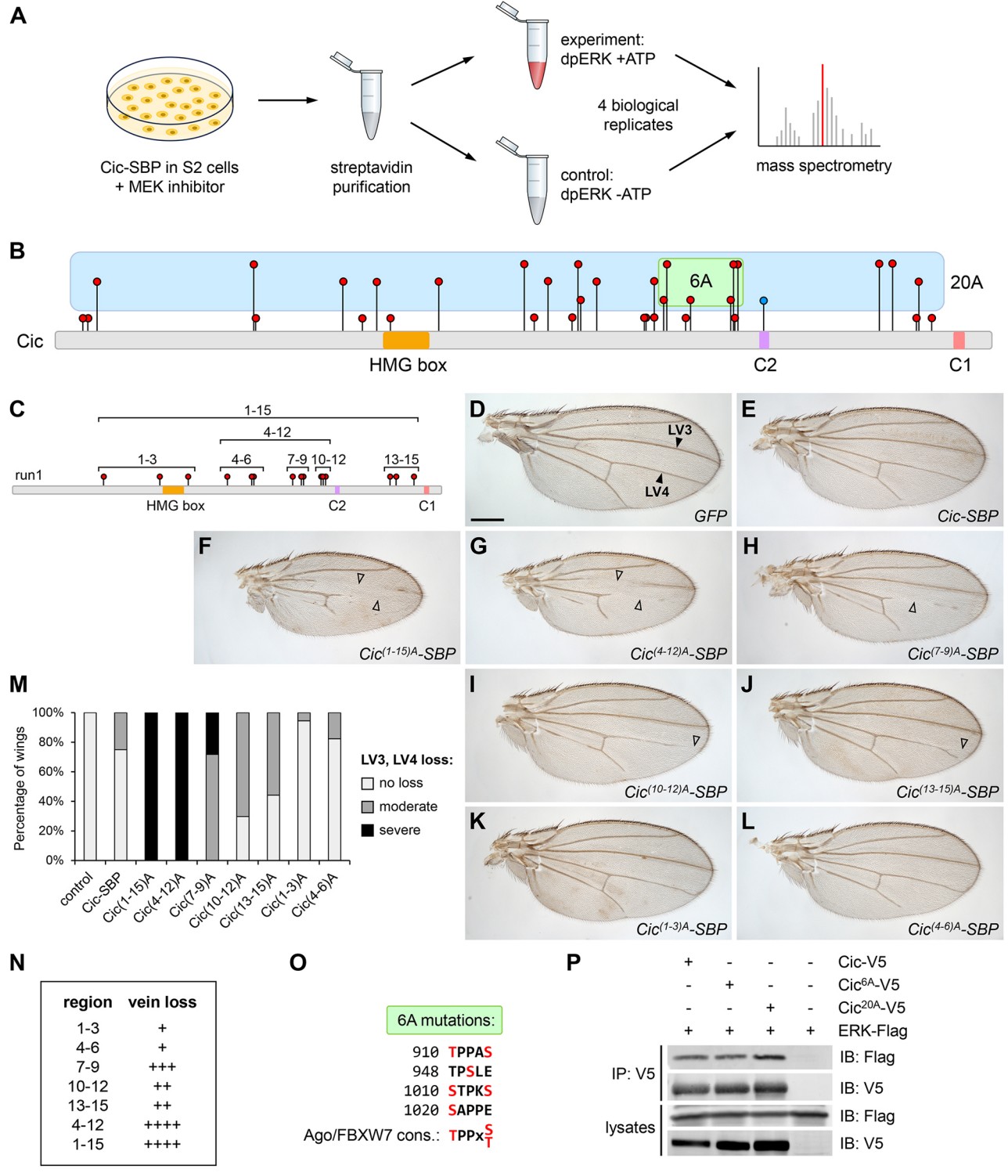

**Fig. 1. ERK-mediated multisite phosphorylation is required for Cic downregulation.** (A) Experimental outline for identification of ERK-mediated phosphorylation sites in Cic. (B) Schematic showing locations and identification frequency of ERK-mediated phosphorylation sites in Cic, based on four biological replicates. The height of site markers is proportional to the identification frequency (between 1 and 4 times). Green and blue shading shows sites chosen for mutagenesis in the Cic[6A] and Cic[20A] constructs, respectively. Individual sites are shown in Figs. S1 and S2. (C) Schematic showing locations and groups of mutated residues in the corresponding *UAS-Cic-SBP* constructs, based on the first mass spectrometry run (run1, 15 sites total). Individual sites are shown in Fig. S1. (D–L) Adult male wing phenotypes resulting from expression of the indicated *Cic-SBP* variants under the control of the wing-specific *MS1096-GAL4* driver. Scale bar: 300 µm. Arrowheads in D point to longitudinal veins 3 and 4 (LV3, LV4). Open arrowheads in F–J indicate vein loss. (M) Quantification of LV3 and LV4 loss in experiments as per D–L. $n \geq 27$ for each genotype. Moderate loss: one incomplete and one complete LV3/4 (e.g. I), severe loss: both LV3 and LV4 incomplete (e.g. F). (N) Summary of the severity of vein loss in experiments as per F–L, with + indicating the weakest phenotype and ++++ indicating the strongest phenotype. (O) Predicted Ago/FBXW7 phosphodegrons in Cic. Phosphorylated residues are highlighted in red, with consensus sequence shown at the bottom. (P) Western blot showing co-immunoprecipitation between V5-tagged Cic variants and Flag-tagged ERK in *Drosophila* S2 cells. Mutating the indicated sites to alanine did not affect the binding. Blot image representative of three independent repeats.

(Fig. 2H,I). At the same time, the protein levels of Cic–mVenus variants progressively increased from lowest for the wild-type Cic (Fig. 2J) to highest for the Cic$^{\Delta C2}$ mutant (Fig. 2M), with intermediate levels for Cic$^{6A}$ and Cic$^{20A}$ (Fig. 2K,L). A similar trend was observed when the Cic$^{6A}$ and Cic$^{20A}$ phosphomutant variants were expressed in the eye, where Cic is also downregulated by EGFR to ERK signaling during development (Tseng et al., 2007). Expression of Cic variants with a progressive increase of phosphosite substitutions using the eye-specific driver *GMR-GAL4* led to a corresponding increase in the severity of eye loss (Fig. 2N–R, quantified in Fig. 2S,T). As in the wing, expression of the Cic$^{\Delta C2}$ mutant gave the strongest phenotype in the eye (Fig. 2Q,T).

Collectively, these results suggest that the Cic$^{20A}$ mutations eliminated most of the phosphosites important for Cic downregulation by ERK. The Cic$^{6A}$ mutations, however, do not account for all of the effects of Cic phosphorylation by ERK,

and additional sites included in Cic$^{20A}$ strongly contribute to downregulation. As the Cic$^{\Delta C2}$ mutant exhibited a further increase in phenotype severity, there might be additional functional phosphosites present in Cic (e.g. those that were identified once in our mass spectrometry analysis) that were not included in Cic$^{20A}$.

## ERK-mediated multisite phosphorylation of Cic is required for Cic degradation and target gene expression

Downregulation of Cic by ERK is well characterized in the early embryo, where spatially restricted activation of Torso induces phosphorylation and activation of ERK, leading to a reduction of Cic levels at both poles of the embryo, presumably through proteolytic degradation (Astigarraga et al., 2007; Grimm et al., 2012). We generated transgenic fly lines carrying *pTIGER*-based mVenus-tagged Cic phosphomutants and expressed them using the maternal driver *MTD-GAL4* to study their effects in the early embryo. The

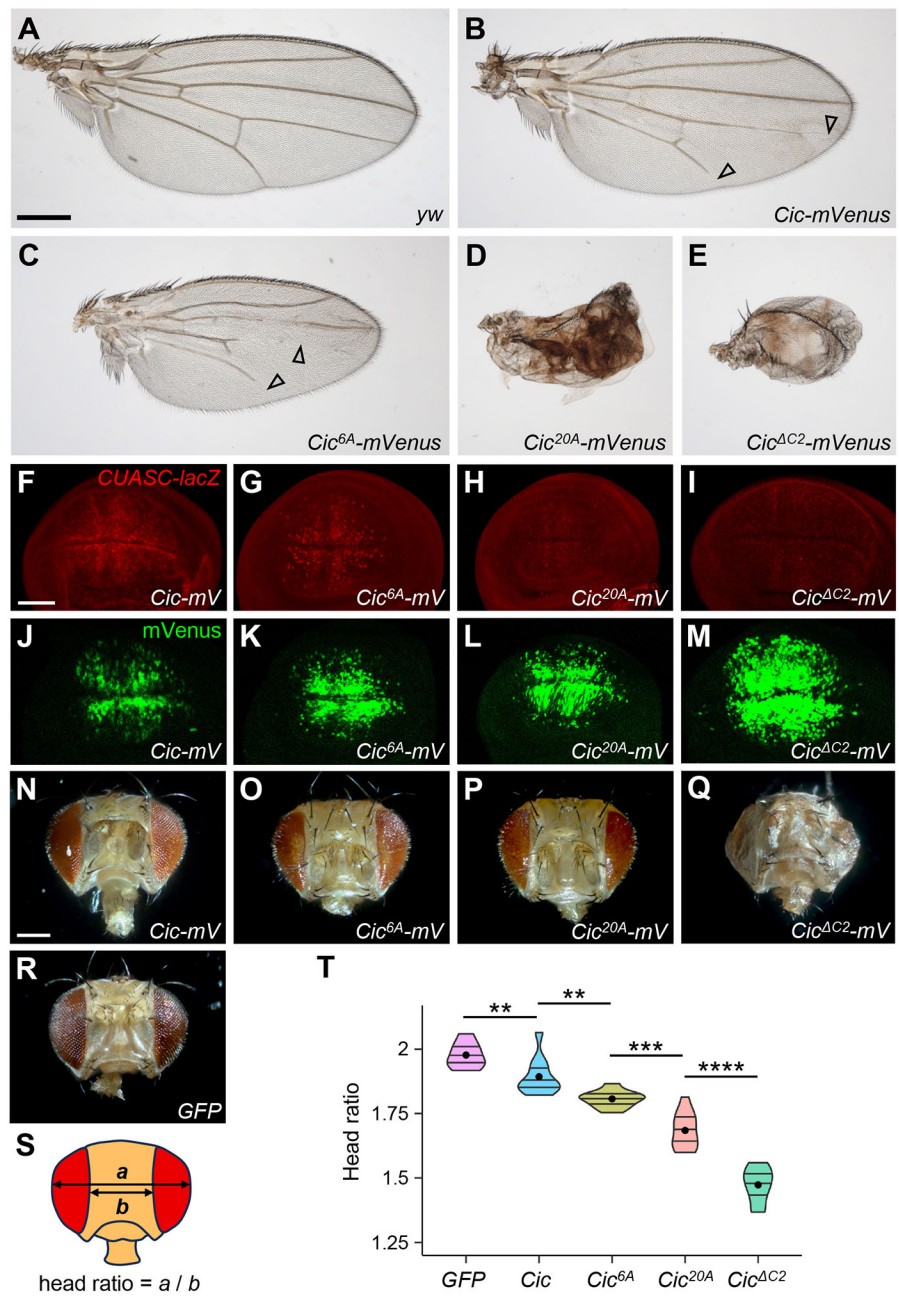

**Fig. 2. Cic activity *in vivo* correlates with the degree of phosphosite substitution.** (A–E) Adult female wing phenotypes resulting from expressing the indicated *Cic-mVenus* variants under the control of the wing-specific *MS1096-GAL4* driver. Open arrowheads indicate vein loss. Images representative of three independent repeats. Scale bar: 300 µm. (F–M) Wing imaginal discs from third instar larvae expressing *CUASC-lacZ* and the indicated *Cic-mVenus* (*mV*) variants under the control of the wing-specific *C5-GAL4* driver. Discs were immunostained for LacZ (F–I) or mounted without staining to visualize mVenus fluorescence (J–M). Images representative of three independent repeats. Scale bar: 50 µm. (N–R) Adult female head phenotypes resulting from expressing the indicated *Cic-mVenus* variants or *GFP* control using the eye-specific *GMR-GAL4* driver. Scale bar: 200 µm. (S,T) Quantification of the head phenotypes shown in N–R, shown as a violin plot with median and quartile indicated by line and mean by the dot. *n*≥10 for each genotype. Statistics calculated by one-way ANOVA (*F*=97.81, *P*<0.001) followed by post-hoc two-tailed unpaired Student's *t*-tests (**P<0.01, ***P<0.001, ****P<0.0001).

wild-type Cic–mVenus was properly downregulated in the cell nuclei at the poles where Torso is active, compared to the higher nuclear levels observed in the middle of the embryo (Fig. 3A–A″). In contrast, the Cic[6A] and the Cic[20A] mutants exhibited prominent nuclear signal at the poles (Fig. 3B–C″) suggesting that these variants are resistant to downregulation by ERK via protein degradation. In this assay, the phenotype for Cic[20A] was indistinguishable from that of Cic[ΔC2] (Fig. 3C–D″). Quantification of Cic protein levels confirmed a significant increase for the Cic[20A] and Cic[ΔC2] proteins in the anterior region of the embryo (Fig. 3K).

At the embryo termini, downregulation of Cic allows for expression of downstream Torso pathway target genes such as *huckebein* (*hkb*) (Jimenez et al., 2000). We studied the expression pattern of *hkb* in embryos with maternally provided expression of Cic phosphomutants using fluorescence *in situ* hybridization (FISH). Wild-type Cic–mVenus slightly decreased the posterior domain of *hkb* expression, and expression of the Cic[6A] or Cic[20A] mutants showed a significant further decrease (Fig. 3E–H,L).

Expression of Cic[ΔC2] resulted in a severe loss of *hkb* expression, with 75% of embryos lacking the signal altogether (Fig. 3I,J,L). These data show that the phosphosites we identified are required for ERK-mediated downregulation and relief of repression that allows target gene expression in the Torso/ERK signaling pathway. As in the other assays, *hkb* expression analysis demonstrated that multisite phosphorylation of Cic by ERK is crucial for achieving the proper level of downregulation.

Higher stability of the Cic[6A] or Cic[20A] mutants *in vivo* suggests that these variants are resistant to proteasomal degradation. To test this more directly, we expressed Cic–mVenus variants in cultured *Drosophila* S2 cells in the presence or absence of the proteasomal inhibitor, MG132 (Tsubuki et al., 1996). In this assay, Cic variants were co-expressed with ERK[Sem], a hyperactive form of ERK (Brunner et al., 1994) that we have previously shown can promote Cic phosphorylation (Paul et al., 2020; Yang et al., 2016). Proteasomal inhibition resulted in a significant increase in the levels of wild-type Cic-mVenus (Fig. 4A,B). In contrast, neither Cic[6A] nor Cic[20A]

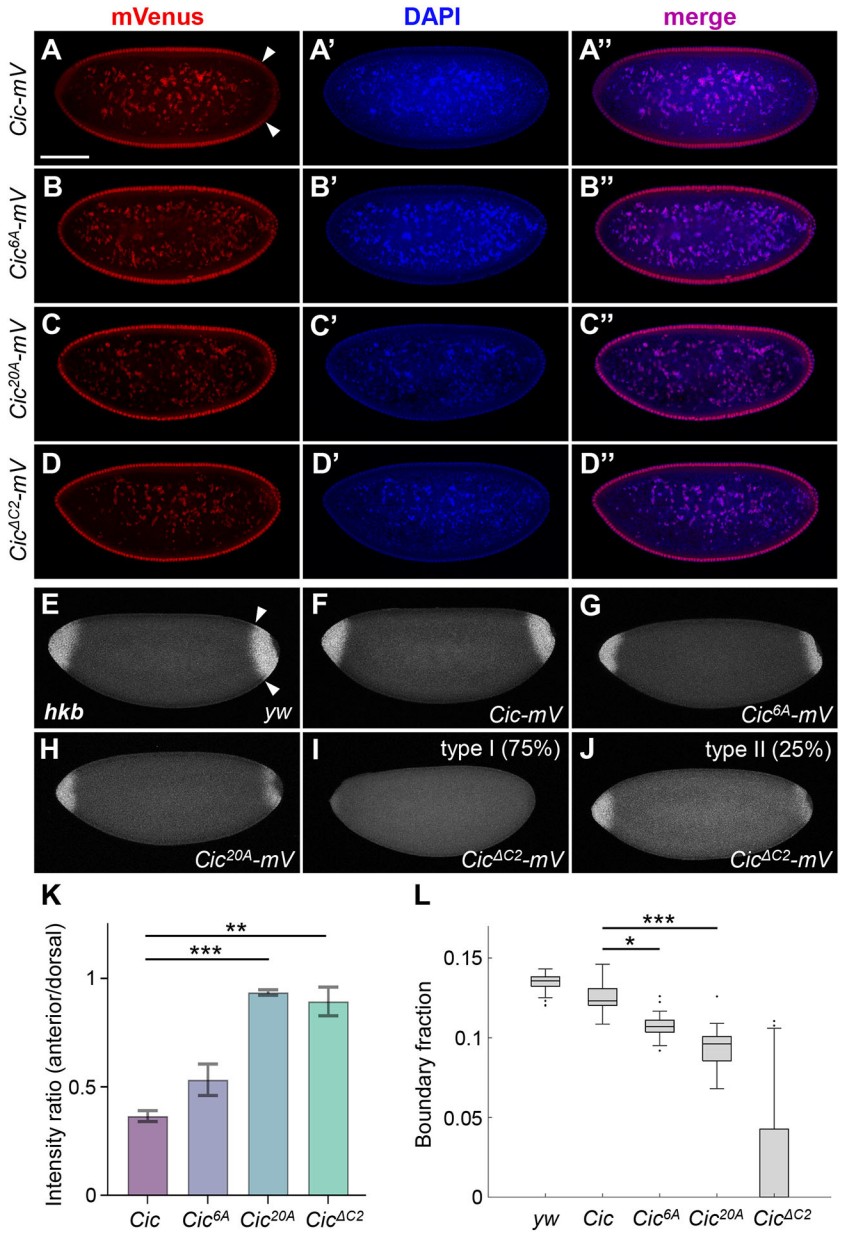

**Fig. 3. ERK-mediated multisite phosphorylation of Cic is required for Cic degradation and target gene expression.** (A–D″) Cic–mVenus immunofluorescence in embryos collected from females expressing the indicated *Cic-mVenus* variants under the control of the maternal *MTD-GAL4* driver. Red, mVenus signal; blue, DAPI (DNA) signal. Scale bar: 100 µm. Arrowheads in A delimit the posterior domain of Cic degradation. (E–J) Fluorescence *in situ* hybridization for *huckebein* (*hkb*) in nuclear cycle 14 embryos collected from females expressing the indicated *Cic-mVenus* variants under the control of the maternal *MTD-GAL4* driver. *hkb* expression was undetectable in the posterior region in 75% of embryos ('type I', 21 out of 28) derived from the *MTD>Cic[ΔC2]* females (I,J). (K) Quantification of Cic–mVenus immunofluorescence shown in A–D. *n*=3 for each genotype. Total intensity in a rectangular box in the anterior region of the embryo was divided by total intensity from the same box (turned by 90°) in the dorsal region. Error bars denote s.e.m. **$P<0.01$, ***$P<0.001$ (two-tailed unpaired Welch's *t*-test). (L) Quantification of *hkb* expression shown in E–J. *n*≥23 for each genotype. The length of the posterior *hkb* expression region was measured relative to the embryo perimeter and represented as boundary fraction. The embryo was segmented using an active contour technique, and the *hkb* expressing region was identified along the boundary contour by its change in intensity relative to non-expressing regions of *hkb* (see Materials and Methods). The box represents the 25–75th percentiles, and the median is indicated. The whiskers show the non-outlier maximum and minimum for each group. The outliers are values which are more than 1.5× IQR (interquartile range) away from the 25th or 75th percentiles. *$P<0.05$, ***$P<0.001$ (multiple comparison Tukey–Kramer test).

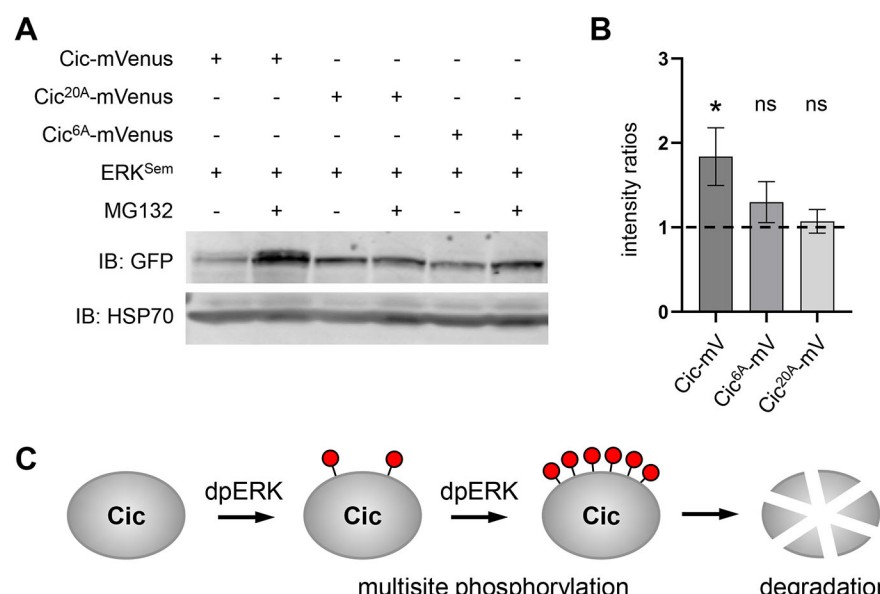

**Fig. 4. The Cic6A and Cic20A variants are resistant to proteasomal degradation.** (A) A representative western blot showing expression levels of the indicated Cic–mVenus variants in protein extracts from S2 cells, with and without MG132. Activated ERK (ERK^Sem) was co-expressed with Cic–mVenus in all samples. HSP70 was used as a loading control. (B) Quantification of the data shown in A, from $n$=3 biological replicates. Protein levels for the Cic variants in the presence of MG132 were divided by the corresponding levels without MG132 within each experimental run, and these ratios were analyzed using one-sample $t$-test against a baseline of 1 (dotted line). Mean ratios are plotted, with error bars denoting s.e.m. *$P$<0.05; ns, not significant. See Materials and Methods for further details. (C) Summary model. ERK-mediated multisite phosphorylation of Cic results in downregulation of Cic activity, at least in part via degradation.

showed a significant increase over their baselines, confirming that they were resistant to proteasomal degradation (Fig. 4A,B).

## Conclusions

In this work, we have identified phosphosites in Cic that are directly targeted by ERK and tested their functions *in vivo* via expression of mutagenized Cic variants. These assays involve developmental processes that are dependent on ERK-mediated downregulation of Cic activity as a transcriptional repressor, such as the EGFR-dependent patterning of the wing veins and the Torso-dependent gene expression in the embryo. Consistently across these assays, the Cic6A, Cic20A and Cic(1-15)A variants behaved as gain-of-function Cic mutants with a stronger repressor activity than wild-type Cic (i.e. as super-repressors), as indicated by the mutant phenotypes. Cic^ΔC2 was the strongest repressor because it impairs binding to ERK and is expected to prevent all ERK-mediated phosphorylation. We note that controlled overexpression like the one used in this study might be the only feasible approach to examine the effects of these mutations *in vivo*, as attempts to introduce them into the *Drosophila* genome using genome editing would likely be unsuccessful, given the observed strong gain-of-function phenotypes. In support of this possibility, previous attempts to mutagenize the C2 domain using CRISPR-Cas9 genome editing only resulted in alleles missing one or two amino acids, with longer deletions presumably causing dominant lethality (Fores et al., 2017a).

Importantly, partial sets of phosphosite mutations gave weaker phenotypes, indicating the requirement for multisite phosphorylation for proper Cic downregulation (Fig. 4C). Our assays suggest that such multisite phosphorylation is necessary for Cic degradation (Fig. 4A–C), although it might also contribute to controlling other aspects of Cic function, such as DNA binding, nucleocytoplasmic shuttling or repressor activity (Astigarraga et al., 2007; Grimm et al., 2012; Keenan et al., 2020; Lim et al., 2013; Rodriguez-Munoz et al., 2022). Consistent with the latter, a recent study has proposed that multisite phosphorylation might result in an intramolecular conformational change in mouse CIC, which leads to its dissociation from the DNA and promotes cytoplasmic localization (Park et al., 2022).

The sites that we mutated within the Cic6A and Cic(7-9)A subsets are similar to the Ago/FBXW7 phosphodegron consensus sequence pTPPxpS/T, where the first phosphorylated threonine is the crucial phosphorylated residue, and the second phosphorylated serine or threonine, also occasionally represented by the negatively charged aspartic acid or glutamic acid, helps in recognition (Singh et al., 2022). These regions in Cic play a substantial role in mediating ERK-dependent downregulation (Fig. 1N); however, other phosphosites are required for the full effect. Although these results are consistent with a report that Ago contributes to Cic degradation in flies (Suisse et al., 2017), further biochemical experiments are needed to ascertain that Ago indeed targets the identified phosphodegron, and it is possible that other ubiquitin ligases participate in Cic proteasomal degradation. Also, we cannot rule out that the identified sites might be targeted *in vivo* by other kinases in addition to ERK.

Phosphodegron-mediated degradation has been shown to be an important mechanism for the regulation of transcriptional activators and repressors in both yeast and mammals (Holt, 2012; Liu et al., 2010). The homology between the human and fly Cic orthologs is low outside the conserved HMG box and C1 domains that mediate DNA binding (Fores et al., 2017b; Webb et al., 2025), making it difficult to align the phosphosites in *Drosophila* Cic with the human CIC sequence. Instead, we have identified six putative phosphodegrons in the C-terminal half of human CIC, in a general location that is similar to where the predicted phosphodegrons are in the fly Cic (Fig. S3). Some of the serine and threonine residues in these motifs were found to be phosphorylated in high-throughput studies (Fig. S3, PhosphoSitePlus). We speculate that human CIC downregulation might occur at least in part via these sequences.

## MATERIALS AND METHODS
### *Drosophila melanogaster* stocks

Fly stocks and crosses were maintained on standard yeast-cornmeal-molasses-agar medium at 25°C or 18°C. The following driver lines were used: *MS1096-GAL4* (wing pouch) (Capdevila and Guerrero, 1994), *GMR-GAL4* (eye) (Hay et al., 1997) and *MTD-GAL4* (maternal triple driver) (Mazzalupo and Cooley, 2006). Transgenic lines were generated by inserting the constructs into the *attP2* genomic site using the φC31-based integration system (Bischof et al., 2007; Venken et al., 2006). Injections were performed by Rainbow Transgenic Flies, Inc. All constructs were integrated in the same site and therefore provide a matching set for comparisons of expression levels and phenotypes.

## Plasmid construction

Construction of C-terminally tagged full length *pMK33-Cic-SBP* and *pUAST-Cic-SBP* was as described previously (Yang et al., 2016). For making Cic[(1-15)A] and Cic[20A], gene fragments with mutations were synthesized by Twist Biosciences. Construction of *pUAST-attB-Cic[(1-15)A]-SBP* and *pUAST-attB-Cic[20A]-mVenus* was carried out by assembling gene fragments using the HiFi DNA assembly Kit (NEB). Construction of the other subgroups of Cic mutants was done by assembling different parts of Cic generated by overlap PCR using wild-type Cic, Cic[(1-15)A], and Cic[20A] as templates. For maternal expression, Cic variants were cloned into pTIGER vector (Ferguson et al., 2012). Construction of C-terminally tagged *ERK-Flag* was described previously (Tipping et al., 2010). *pMT-ERK[Sem]-Flag* was described previously (Yang et al., 2016; Paul et al., 2020). For co-immunoprecipitation and proteasomal degradation experiments, wild-type Cic and mutant variants tagged with mVenus or V5 were subcloned into pMT/V5-His vectors (Invitrogen).

## *In vitro* kinase reactions

For stable expression in S2 cells (a gift from Spyros Artavanis-Tsakonas, Harvard Medical School, USA), the pMK33-Cic-SBP construct was transfected by using Effectene transfection reagent (Qiagen), and stable cell lines were selected in the presence of 300 µg/ml hygromycin (Sigma), as described previously (Yang and Veraksa, 2017). *pMK33-Cic-SBP* stable cells were pre-incubated with 2 µM PD0325901, a MEK inhibitor (Biotang Inc.), with DMSO as vehicle, for 3 h before induction. Cells were induced with 0.35 mM CuSO$_4$ overnight. Cells were harvested and then lysed with default lysis buffer (50 mM Tris-HCl pH 7.5, 125 mM NaCl, 5% glycerol, 0.2% IGEPAL CA-630, 1.5 mM MgCl$_2$, 1 mM DTT, 25 mM NaF, 1 mM Na$_3$VO$_4$ and 1 mM EDTA) containing 2× Complete protease inhibitor (Roche). Cleared cell lysates were incubated with Streptavidin beads (Pierce) at 4°C for 2 h. After three washes (with the final wash in kinase buffer), 500 ng of purified dpERK and 200 mM ATP in kinase buffer (Cell Signaling Technologies) were added and samples were incubated at 30°C for 30 min. dpERK without ATP was used as a negative control. Purification of phosphorylated ERK from bacteria was described previously (Paul et al., 2020). After several washes, samples were eluted with 2× SDS sample buffer, analyzed on 6% SDS-PAGE gels, and Cic phosphorylation was analyzed by nanoLC-MS/MS at the Taplin Mass Spectrometry Facility at Harvard Medical School.

## Identification of phosphorylation sites by mass spectrometry

Excised gel bands were cut into approximately 1 mm$^3$ pieces. The samples were reduced with 1 mM DTT for 30 min at 60°C and alkylated with 5 mM iodoacetamide for 15 min in the dark at room temperature. Gel pieces were then subjected to a modified in-gel trypsin digestion procedure. Gel pieces were washed and dehydrated with acetonitrile for 10 min followed by removal of acetonitrile. Pieces were then completely dried in a speed-vac. Rehydration of the gel pieces was with 50 mM ammonium bicarbonate solution containing 12.5 ng/µl modified sequencing-grade trypsin (Promega, Madison, WI, USA) at 4°C. Samples were then placed in a 37°C room overnight. Peptides were later extracted by removing the ammonium bicarbonate solution, followed by one wash with a solution containing 50% acetonitrile and 1% formic acid. The extracts were then dried in a speed-vac (∼1 h). The samples were then stored at 4°C until analysis.

On the day of analysis, the samples were reconstituted in 5–10 µl of HPLC solvent A (2.5% acetonitrile, 0.1% formic acid). A nano-scale reverse-phase HPLC capillary column was created by packing 2.6 µm C18 spherical silica beads into a fused silica capillary (100 µm inner diameter×∼30 cm length) with a flame-drawn tip. After equilibrating the column each sample was loaded via a Famos auto sampler (LC Packings, San Francisco CA) onto the column. A gradient was formed, and peptides were eluted with increasing concentrations of solvent B (97.5% acetonitrile, 0.1% formic acid).

As each peptide was eluted, they were subjected to electrospray ionization and then they entered into an LTQ Orbitrap Velos Pro ion-trap mass spectrometer (Thermo Fisher Scientific). Eluting peptides were detected, isolated and fragmented to produce a tandem mass spectrum of specific fragment ions for each peptide. Peptide sequences (and hence protein identity) were determined by matching protein or translated nucleotide databases with the acquired fragmentation pattern by the software program, Sequest (ThermoFinnigan, San Jose, CA, USA; Eng et al., 1994). The modification of 79.9663 mass units to serine, threonine and tyrosine was included in the database searches to determine phosphopeptides. Phosphorylation assignments were determined by the Ascore algorithm (Beausoleil et al., 2006). All databases include a reversed version of all the sequences and the data was filtered to between a one and two percent peptide false discovery rate. Ascore output is provided as Table S1.

## Co-immunoprecipitation and western blotting

Wild-type and mutant Cic-V5 variants and ERK–Flag were expressed from the pMT vector-based constructs in cultured *Drosophila* S2 cells. S2 cells were cultured at 25°C in standard Schneider's S2 medium with 10% FBS (Gibco) and 5% Pen-Strep (Invitrogen). Proteins were induced with 0.35 mM CuSO$_4$ overnight, cells were lysed in default lysis buffer as above and protein complexes were isolated using anti-V5 beads (A7345, Sigma). After washes and elution with 4× SDS sample buffer, protein complexes were resolved on 7% SDS protein gels and transferred onto Millipore Immobilon-FL PVDF Transfer Membranes with 0.45 µm pores. Primary antibodies used for western blots were as follows: rabbit anti-Flag 1:1000 (F7425, Sigma), mouse anti-V5 1:1000 (V8012, Sigma), rabbit anti-GFP 1:1000 (A11122, Thermo Fisher Scientific), mouse anti-HSP70 1:1000 (H5147, Sigma). Secondary antibodies used were as follows: IRDye 800CW donkey anti-rabbit IgG 1:10,000 (LI-COR) and IRDye 680CW goat anti-mouse IgG, 1:10,000 (LI-COR).

## Immunohistochemistry

To assess embryonic Cic–mVenus localization, 0–4 h embryos collected from *MTD>yw*, *MTD>Cic-mVenus*, *MTD>Cic[6A]-mVenus*, *MTD>Cic[20A]-mVenus* and *MTD>Cic[ΔC2]-mVenus* mothers were dechorionated with 50% (v/v) Clorox bleach, rinsed with water, then fixed for 20 min at room temperature (RT) in a fixative containing 5 ml of 4% (v/v) paraformaldehyde (Electron Microscopy Sciences) in 1× PBS and an equal volume of heptane. After fixation, the embryos were devitellinized via the addition of 8 ml of methanol and harsh agitation for 90 s. Fixed and devitellinized embryos were collected, washed three times in methanol and four times in ethanol, then stored in ethanol at −20°C. Embryos were rehydrated once with ethanol, twice with 1:1 ethanol:PBT [1× PBS with 0.1% (v/v) Tween 20], then twice with 1× PBT. The embryos were incubated in blocking reagent [1:1 (v/v) Roche Blocking Reagent and 1× PBT] for 2 h at RT and incubated overnight at 4°C in primary antibody solution [1:100 rabbit anti-GFP (A11122, Thermo Fisher Scientific) in blocking reagent]. Embryos were washed at RT in 0.1% BSA (w/v in 1× PBT), blocked for 1 h then incubated with secondary antibody solution [1:500 goat anti-rabbit IgG conjugated to Alexa Fluor 555 (Thermo Fisher Scientific) in blocking reagent] for 2 h in the dark at RT. Embryos were then washed in the dark then mounted in Prolong Gold Antifade Mountant with DAPI (Thermo Fisher Scientific).

Wing disc staining was performed essentially as described previously (Yang et al., 2016). *UAS-Cic-mVenus* variants were crossed with a line expressing *CUASC-lacZ* under the control of the wing-specific *C5-GAL4* driver (Yang et al., 2016). To visualize Cic expression in wing discs, mVenus-tagged Cic variants were detected directly by mVenus fluorescence after fixation. To visualize LacZ expression, discs were stained with mouse anti-β-galactosidase (LacZ) at 1:100 (Z3783, Promega), followed by incubation with goat anti-mouse IgG Alexa Fluor 555-conjugated secondary antibody (Thermo Fisher Scientific). Discs were mounted and imaged using identical acquisition settings across genotypes. Images were acquired with the Zeiss LSM 880 confocal microscope.

## *hkb* FISH and image analysis

0–4 h embryos were collected as above and used for FISH experiments using standard FISH protocols (Goyal et al., 2017). In brief, ∼50 µl of fixed embryos were incubated in 90% xylenes for 1 h, followed by wash steps with ethanol, methanol and PBT. Embryos were incubated at 65°C with hybridization buffer (50% formamide, 5× SSC, 100 µg/ml sonicated salmon sperm DNA, 50 µg/ml heparin and 0.1% Tween 20) for 4 h. The samples were resuspended with digoxigenin (DIG)-labeled antisense *hkb* RNA probes with hybridization buffer (1:25) and incubated at 65°C overnight.

Journal of Cell Science

After hybridization, the samples were washed with hybridization buffer and PBT, followed by standard immunostaining protocols. DIG-labeled antisense *hkb* RNA probe was synthesized by amplification of the *hkb* cDNA. Nuclear cycle 14 embryos were selected for imaging. DAPI was used for staining nuclei. Sheep anti-DIG (1:25; 11093274910, Roche) was used as primary antibody and Alexa Fluor 568 conjugate (1:500; A-21099, Invitrogen) was used as secondary antibody. Imaging for FISH experiments was performed on a Leica SP5 confocal microscope with following specifications: 20x AIR objective, 405-nm and 561-nm diode lasers.

Segmentation of the *Drosophila* embryo perimeter was performed using the following procedure. From the *hkb*-labeled image, Otsu's method was used to approximately separate the interior and exterior of the embryo. This was followed by a flood fill operation to fill holes in the mask and a morphological dilation. This binary mask was used as input along with the raw image to an active contour technique to finetune the segmentation of the interior of the embryo. The boundary of this mask, the perimeter of the embryo, was isolated as a piecewise parametric curve which was smoothed using a Savitzky–Golay filter. We quantified *hkb* intensity at the embryo border by averaging pixel intensity values within a Euclidean distance of 20 pixels (12.61 µm) of the parametric boundary curve and in the interior of the embryo. Finally, we used MATLAB signal processing toolbox functions 'risetime' and 'falltime' to determine the position on the curve where the *hkb* pole starts and stops. These functions estimate the time instant of a state transition within a signal. Code is available at the GitHub repository: https://github.com/ddenberg/HKB-Quantification.

### Wing and head phenotypes

Adult wings and heads were imaged with Olympus BX60 compound microscope using bright-field illumination and a 4× objective. For the SBP-tagged Cic$^{(1\text{-}15)A}$ series mutants, wings from male progeny of crosses with *MS1096-GAL4* were analyzed. For the mVenus-tagged Cic$^{6A}$, Cic$^{20A}$, and Cic$^{\Delta C2}$ mutants, wings and heads from female progeny from the crosses with *MS1096-GAL4* or *GMR-GAL4* were analyzed. Data visualizations for adult head measurements were generated using the ggpubr (v0.6.0; https://CRAN.R-project.org/package=ggpubr) and ggplot2 (v3.5.2; https://ggplot2.tidyverse.org) packages in R (v4.4.1). To compare groups, violin plots with quantiles and means were generated using the geom_violin() ggplot2 function. The *P*-values from the statistical analyses were added to the plots using the stat_compare_means() ggpubr function. The plot themes were customized using the labs() ggplot2 function.

### Cic proteasomal degradation

*Drosophila* S2 cells were transfected with wild-type or mutant *Cic-mVenus* constructs cloned into the pMT/V5-His expression vectors (see above). Protein expression was induced overnight by adding CuSO$_4$ to a final concentration of 0.35 mM. For proteasome inhibition, cells were treated with MG132 (Sigma) in DMSO at a final concentration of 50 µM for 4 h at 25°C. Control cells were treated with an equivalent volume of DMSO. Cells were harvested, washed once with cold PBS and lysed in default lysis buffer (50 mM Tris-HCl pH 7.5, 125 mM NaCl, 5% glycerol, 0.4% IGEPAL, 1.5 mM MgCl$_2$, 1 mM DTT, 25 mM NaF, 1 mM Na$_3$VO$_4$, 1 mM EDTA and 2× Complete protease inhibitor, Roche). Lysates were clarified by centrifugation at 15,000 *g* for 15 min at 4°C. Supernatants were mixed with SDS sample buffer and boiled for 5 min. Protein samples were analyzed by western blotting (as above).

The experiment was performed in three biological replicates. For statistical analysis of changes after MG132 addition, a one-sample *t*-test was used (Bang et al., 2022). Our null hypothesis was that there was no increase in Cic level after exposure to MG132, meaning that the ratio of band intensities for +MG132 to -MG132 samples would be 1. These ratios were calculated within each western blot, and then used to compute one-sample *t*-tests for each Cic variant against the baseline of 1. After calculating the *t*-statistics, *P*-values were derived using the T.DIST.RT function in Excel. *P*<0.05 was considered significant.

### Acknowledgements

We thank the Bloomington *Drosophila* Stock Center for their services and Roza Khalifa for help with certain experiments. Mass spectrometry was performed at the Taplin Mass Spectrometry Facility at Harvard Medical School.

### Competing interests

The authors declare no competing or financial interests. S.P. is currently employed by Genentech, Inc., South San Francisco, CA, USA, and N.S. is currently employed by Tevard Biosciences, Inc., Boston, MA, USA. Their contributions to this work were made while they were graduate students at the University of Massachusetts Boston.

### Author contributions

Conceptualization: S.P., S.Y.S., A.V.; Data curation: S.P., K.I., N.S., L.Y., D.W.D., W.C., V.L., A.V.; Formal analysis: S.P., K.I., N.S., L.Y., D.W.D., V.L., A.V.; Funding acquisition: S.Y.S., A.V.; Investigation: K.I., N.S., W.C., V.L.; Methodology: S.P., K.I., N.S., L.Y., S.Y.S., A.V.; Project administration: S.Y.S., A.V.; Resources: S.Y.S., A.V.; Software: N.S., D.W.D.; Supervision: S.Y.S., A.V.; Validation: S.P., K.I., N.S., A.V.; Visualization: S.P., K.I., N.S., L.Y., D.W.D., W.C., V.L., A.V.; Writing – original draft: S.P., A.V.; Writing – review & editing: K.I., N.S., L.Y., S.Y.S., A.V.

### Funding

This work was supported by the National Institute of Health grants GM141843 to A.V. and S.Y.S., and GM158116 to A.V. S.Y.S. acknowledges support from the Princeton Catalysis Initiative. S.P. was supported by the Sanofi and Oracle UMass Boston Doctoral Fellowships. Open Access funding provided by University of Massachusetts Boston. Deposited in PMC for immediate release.

### Data and resource availability

The raw mass spectrometry proteomics data have been deposited to the ProteomeXchange Consortium via the PRIDE (Perez-Riverol et al., 2025) partner repository with the dataset identifier PXD066179. Code for quantifying embryo FISH data is available at the GitHub repository: https://github.com/ddenberg/HKB-Quantification. All relevant data and details of resources can be found within the article and its supplementary information.

### Peer review history

The peer review history is available online at https://journals.biologists.com/jcs/lookup/doi/10.1242/jcs.264327.reviewer-comments.pdf

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
