## [Peer Review File · Journal of Cell Science]

ERK inhibits Capicua repressor function via multisite phosphorylation

Sayantanee Paul, Khandan Ilkhani, Nathan Strozewski, Liu Yang, David W. Denberg, Wootchelmine Christalin, Vanessa Locke, Stanislav Y. Shvartsman and Alexey Veraksa
DOI: 10.1242/jcs.264327

Editor: Simon Cook

Review timeline

Original submission:	25 July 2025
Editorial decision:	10 September 2025
First revision received:	18 January 2026
Editorial decision:	16 February 2026
Second revision received:	17 February 2026
Accepted:	23 February 2026

Original submission

First decision letter

MS ID#: jcs.264327

MS TITLE: ERK inhibits Capicua repressor function via multisite phosphorylation

AUTHORS: Sayantanee Paul; Khandan Ilkhani; Nathan Strozewski; Liu Yang; David W Denberg; Wootchelmine Christalin; Vanessa Locke; Stanislav Y Shvartsman; Alexey Veraksa

ARTICLE TYPE: Short Report

Dear Dr Veraksa,

We have now reached a decision on the above manuscript.

To see the reviewers' reports and a copy of this decision letter, please go to:

As you will see, the reviewers raise a number of substantial criticisms that prevent me from accepting the paper at this stage. They suggest, however, that a revised version might prove acceptable, if you can address their concerns. If you think that you can deal satisfactorily with the criticisms on revision, I would be pleased to see a revised manuscript. We would then return it to the reviewers.

I would draw your attention especially to one of the referees comments "the hypothesis that ERK-mediated phosphorylation directly promotes CIC turnover, data supporting this is somewhat weak in the current manuscript" This is an area which I feel needs to be strengthened. In particular, how many phosphorylation sites are direct ERK1/2 sites (S-P or T-P) and which of these are critical for CIC turnover? Or is ERK driving activation of an ERK-dependent kinase which is responsible for the phosphorylation sites that mediate CIC turnover.

Reviewer 1

Advance summary and potential significance to field

Capicua (CIC) is a transcriptional repressor essential for developmental patterning and tumor suppression, whose function is modulated by ERK/MAPK signaling. ERK-dependent phosphorylation has been suggested to regulate CIC nuclear localization and/or stability through targeted degradation, thereby relieving repression of growth and differentiation genes. This dynamic control ensures proper developmental transitions but, when dysregulated, contributes to cancer progression by enabling aberrant gene expression programs. However, the precise ERK-mediated phosphorylation sites governing CIC regulation remain poorly defined, and their detailed characterization is critical for understanding both normal development and CIC-driven pathologies. To address this question, the authors purified *Drosophila* CIC and incubated it with recombinant ERK *in vitro*, followed by mass spectrometry-based identification of phosphorylation sites. This approach revealed more than 20 potential ERK targets. Through a series of combinatorial mutations and *in vivo* expression of CIC phospho-mutants, the authors assessed how site-specific phosphorylation influences CIC stability as well as its developmental functions in wing vein formation and embryonic patterning. This revealed that ERK needs to phosphorylate CIC on multiple sites to maximally inactivate it. Examination of CIC levels in the embryo show that the phospho mutants lead to CIC stabilisation, suggesting ERK phosphorylation of CIC promotes its degradation. The authors postulate that this multisite phosphorylation likely targets phosphodegrons that are recognized by ubiquitin ligases such as Ago/FBXW7, contributing to CIC degradation. The work is rigorous and provides a comprehensive framework for dissecting how ERK-mediated phosphorylation modulates CIC function. By systematically mapping and functionally testing candidate phosphorylation sites, the authors not only clarify mechanisms of CIC regulation but also highlight the broader principle of how signaling-dependent post-translational modifications fine-tune transcriptional repressors during development. These insights lay the foundation for future studies aimed at linking CIC phosphorylation dynamics to disease contexts, including cancer, where aberrant ERK activity and CIC dysfunction frequently intersect.

Major Point

- Results Section Organization. The flow of the Results section could be improved for clarity and readability. Currently, the authors present phosphorylation analysis and follow-up *in vivo* experiments with UAS transgenes based on a single mass spectrometry run (Figure 1). They then repeat the mass spectrometry analysis three additional times, identify further putative sites, and generate new transgenes (Figure 2).

This division feels somewhat disjointed: Figure 1 concludes with mutations that yield phenotypes, but Figure 2 begins by essentially repeating the mass spectrometry analysis. For coherence, these data should ideally be presented together as a single, consolidated section and figure, incorporating the compiled mass spectrometry results and subsequent experimental validation. This restructuring would reduce redundancy, improve the narrative flow, and make the overall results easier for readers to follow.

- CIC levels in wing vein precursors. The adult wing vein experiments are clear and convincing. However, the study would be strengthened by examining CIC levels (via the Venus tag) in L3 wing imaginal discs. At this stage, CIC is normally downregulated in future vein regions in cells with high dpERK. Demonstrating whether the phospho-mutants escape this downregulation would nicely complement the vein phenotypes and embryonic data, and further support the narrative that ERK-mediated phosphorylation drives CIC degradation.

- CIC degradation. The manuscript would be significantly strengthened by more robust evidence that the phospho-mutants are protected from degradation. A straightforward approach would be to express the mutants in S2 cells with and without an ERK activator (such as RAS-V12) and assess whether CIC phospho-mutants are resistant to degradation, and to what extent. Performing the experiment in the presence and absence of a proteasome inhibitor would further demonstrate that the observed effects are due to degradation. This would strengthen the hypothesis that ERK-mediated phosphorylation directly promotes CIC turnover, data supporting this is somewhat weak in the current manuscript. Such an assay could also be extended to test binding with Ago, providing a direct readout of the proposed phospho-degron mechanism. Without additional experiments probing the mechanism of action of the phosphorylation events, the manuscript remains limited in terms of advancing our understanding of how ERK-mediated phosphorylation regulates CIC stability.

Minor points.

- It is worth acknowledging in the discussion that the *in vivo* experiments do not directly prove ERK-mediated phosphorylation, as other kinases could also contribute to Cic regulation *in vivo*. Explicitly stating this limitation would provide a more balanced interpretation of the findings.

Reviewer 2

Advance summary and potential significance to field

The Capicua (Cic) transcriptional repressor controls multiple developmental processes and has a well-established role as a target of Erk signalling. Cic is directly phosphorylated by Erk, which suppresses its repressive activity, possibly through a variety of mechanisms. A major Erk docking site has been previously identified in Cic, but the exact complement of sites targeted by Erk phosphorylation, and how they lead to Cic downregulation, remains unknown. Here, the authors identify a set of 21 Erk-mediated phosphorylation sites on Cic, which exert a cumulative inhibitory influence on its activity. Mutation of 20 sites -the remaining site maps to the Erk docking site and is therefore difficult to analyse- induces gain-of-function activities of Cic in the embryo and the adult, indicating that those sites contribute a substantial portion of Erk's inhibitory effect. (However, they do not explain the entire effect, as removal of the Erk docking site -and thus, presumably, complete abrogation of Cic phosphorylation- results in stronger phenotypes.) This work therefore represents a further step in our understanding of Cic regulation via Erk signalling. The paper is clearly written and easy to follow, with results carefully documented in high-quality figures.

Comments for the author

Major comments [Please request additional experiments only if they are essential for supporting the conclusions; authors should be encouraged to highlight any claims that are preliminary or speculative, or to discuss any pitfalls or alternative interpretations in a 'Limitations' section]

One suggestion for improvement would be to reconsider the emphasis given in the Conclusion to the notion that the dominant phosphorylation sites probably function as mediators of Cic degradation by Archipelago (Ago), the *Drosophila* FBXW7 ortholog. I understand that this relatively straightforward idea, partially supported by previous findings, helps structure the discussion. However, there is a risk of overemphasizing this model without sufficient evidence, potentially overlooking other mechanisms that may be equally or even more relevant. To begin with, the authors claim that several of the identified sites resemble the FBXW7 phosphodegron, but it remains unclear whether these sites can indeed be recognized by this regulator. Also, the role of Ago in Cic degradation has not been thoroughly tested. For example, it has been reported that knockdown of maternal ago transcripts causes a partial embryonic pair-rule phenotype, rather than the terminal defects associated with Erk or Torso receptor inactivation (Staller et al., 2013). Ago may not have been fully inactivated in those experiments; nonetheless, I think its potential regulatory role on Cic requires further analysis, at least in the embryo. Moreover, Cic phosphorylation must trigger additional regulatory mechanisms that are either independent of, or likely precede, its degradation, as observed, for example, in the follicular epithelium of the ovary and embryonic neuroectoderm (Rodríguez-Muñoz et al. 2022; Lim et al. 2013). The authors briefly mention these mechanisms but, in my opinion, the overall perspective feels somewhat unbalanced.

Minor comments

The Cic C1 domain is mentioned in the Conclusion and highlighted in Fig. S2, but without any explanation of what it is. I suggest writing "...the conserved HMG box and C1 DNA binding domains..." (p. 9) and including the corresponding reference (Fores et al., 2017).

First revision

Author response to reviewers' comments

Response to Reviewers' comments for the manuscript Paul et al., jcs.264327, titled "ERK inhibits Capicua repressor function via multisite phosphorylation". Reviewers' comments are italicized.

Reviewer 1:

Capicua (CIC) is a transcriptional repressor essential for developmental patterning and tumor suppression, whose function is modulated by ERK/MAPK signaling. ERK- dependent phosphorylation has been suggested to regulate CIC nuclear localization and/or stability through targeted degradation, thereby relieving repression of growth and differentiation genes. This dynamic control ensures proper developmental transitions but, when dysregulated, contributes to cancer progression by enabling aberrant gene expression programs. However, the precise ERK-mediated phosphorylation sites governing CIC regulation remain poorly defined, and their detailed characterization is critical for understanding both normal development and CIC-driven pathologies. To address this question, the authors purified Drosophila CIC and incubated it with recombinant ERK in vitro, followed by mass spectrometry-based identification of phosphorylation sites. This approach revealed more than 20 potential ERK targets. Through a series of combinatorial mutations and in vivo expression of CIC phospho-mutants, the authors assessed how site-specific phosphorylation influences CIC stability as well as its developmental functions in wing vein formation and embryonic patterning. This revealed that ERK needs to phosphorylate CIC on multiple sites to maximally inactivate it. Examination of CIC levels in the embryo show that the phospho mutants lead to CIC stabilisation, suggesting ERK phosphorylation of CIC promotes its degradation. The authors postulate that this multisite phosphorylation likely targets phosphodegrons that are recognized by ubiquitin ligases such as Ago/FBXW7, contributing to CIC degradation. The work is rigorous and provides a comprehensive framework for dissecting how ERK-mediated phosphorylation modulates CIC function. By systematically mapping and functionally testing candidate phosphorylation sites, the authors not only clarify mechanisms of CIC regulation but also highlight the broader principle of how signaling-dependent post-translational modifications fine-tune transcriptional repressors during development. These insights lay the foundation for future studies aimed at linking CIC phosphorylation dynamics to disease contexts, including cancer, where aberrant ERK activity and CIC dysfunction frequently intersect.

Major Point

- Results Section Organization. The flow of the Results section could be improved for clarity and readability. Currently, the authors present phosphorylation analysis and follow-up in vivo experiments with UAS transgenes based on a single mass spectrometry run (Figure 1). They then repeat the mass spectrometry analysis three additional times, identify further putative sites, and generate new transgenes (Figure 2).

This division feels somewhat disjointed: Figure 1 concludes with mutations that yield phenotypes, but Figure 2 begins by essentially repeating the mass spectrometry analysis. For coherence, these data should ideally be presented together as a single, consolidated section and figure, incorporating the compiled mass spectrometry results and subsequent experimental validation. This restructuring would reduce redundancy, improve the narrative flow, and make the overall results easier for readers to follow.

We thank the Reviewer for this suggestion and agree that this reorganization would improve the flow. Accordingly, previous Figures 1 and 2 were merged into the new Figure 1, with corresponding updates in the text.

- CIC levels in wing vein precursors. The adult wing vein experiments are clear and convincing. However, the study would be strengthened by examining CIC levels (via the Venus tag) in L3 wing

imaginal discs. At this stage, Cic is normally downregulated in future vein regions in cells with high dpERK. Demonstrating whether the phospho- mutants escape this downregulation would nicely complement the vein phenotypes and embryonic data, and further support the narrative that ERK-mediated phosphorylation drives Cic degradation.

We have performed this experiment and also tested the effects of Cic variant overexpression on the provein RTK/ERK activity reporter, CUASC-lacZ. The outcomes were consistent with the other results: CUASC-lacZ expression was progressively decreased, and the levels of Cic-mVenus proteins were increased, as the number of phosphosite mutations increased. These new data are shown in the new Fig. 2, F-M.

- Cic degradation. The manuscript would be significantly strengthened by more robust evidence that the phospho-mutants are protected from degradation. A straightforward approach would be to express the mutants in S2 cells with and without an ERK activator (such as RAS-V12) and assess whether Cic phospho-mutants are resistant to degradation, and to what extent. Performing the experiment in the presence and absence of a proteasome inhibitor would further demonstrate that the observed effects are due to degradation. This would strengthen the hypothesis that ERK-mediated phosphorylation directly promotes Cic turnover, data supporting this is somewhat weak in the current manuscript. Such an assay could also be extended to test binding with Ago, providing a direct readout of the proposed phospho-degron mechanism. Without additional experiments probing the mechanism of action of the phosphorylation events, the manuscript remains limited in terms of advancing our understanding of how ERK-mediated phosphorylation regulates Cic stability.

This experiment proved technically challenging, but we were able to carry it out and obtained important new data. To provide consistently high level of ERK activity, we coexpressed Cic-mVenus variants in S2 cells together with an activated form of ERK, ERK^{Sem}. Under these conditions, proteasome inhibition with MG132 significantly increased the steady-state level of wild-type Cic-mVenus, but not that of Cic^{6A} or Cic^{20A}. This result further solidifies our finding that multisite phosphorylation acts at least in part via Cic degradation. These data are shown in the new Fig. 4A,B.

Minor points.

- It is worth acknowledging in the discussion that the in vivo experiments do not directly prove ERK-mediated phosphorylation, as other kinases could also contribute to Cic regulation in vivo. Explicitly stating this limitation would provide a more balanced interpretation of the findings.

We have added a statement in Conclusion on p. 10 that reads, “Also, we cannot rule out that the identified sites may be targeted in vivo by other kinases in addition to ERK.”

Reviewer 2:

The Capicua (Cic) transcriptional repressor controls multiple developmental processes and has a well-established role as a target of Erk signalling. Cic is directly phosphorylated by Erk, which suppresses its repressive activity, possibly through a variety of mechanisms. A major Erk docking site has been previously identified in Cic, but the exact complement of sites targeted by Erk phosphorylation, and how they lead to Cic downregulation, remains unknown. Here, the authors identify a set of 21 Erk-mediated phosphorylation sites on Cic, which exert a cumulative inhibitory influence on its activity. Mutation of 20 sites -the remaining site maps to the Erk docking site and is therefore difficult to analyse- induces gain-of-function activities of Cic in the embryo and the adult, indicating that those sites contribute a substantial portion of Erk's inhibitory effect. (However, they do not explain the entire effect, as removal of the Erk docking site -and thus, presumably, complete abrogation of Cic phosphorylation- results in stronger phenotypes.) This work therefore represents a further step in our understanding of Cic regulation via Erk signalling. The paper is clearly written and easy to follow, with results carefully documented in high-quality figures.

Major comments

One suggestion for improvement would be to reconsider the emphasis given in the Conclusion to the notion that the dominant phosphorylation sites probably function as mediators of Cic degradation by Archipelago (Ago), the Drosophila FBXW7 ortholog. I understand that this relatively straightforward idea, partially supported by previous findings, helps structure the discussion. However, there is a risk of overemphasizing this model without sufficient evidence, potentially overlooking other mechanisms that may be equally or even more relevant. To begin with, the authors claim that several of the identified sites resemble the FBXW7 phosphodegron, but it remains unclear whether these sites can indeed be recognized by this regulator. Also, the role of Ago in Cic degradation has not been thoroughly tested. For example, it has been reported that knockdown of maternal ago transcripts causes a partial embryonic pair-rule phenotype, rather than the terminal defects associated with Erk or Torso receptor inactivation (Staller et al., 2013). Ago may not have been fully inactivated in those experiments; nonetheless, I think its potential regulatory role on Cic requires further analysis, at least in the embryo. Moreover, Cic phosphorylation must trigger additional regulatory mechanisms that are either independent of, or likely precede, its degradation, as observed, for example, in the follicular epithelium of the ovary and embryonic neuroectoderm (Rodríguez-Muñoz et al. 2022; Lim et al. 2013). The authors briefly mention these mechanisms but, in my opinion, the overall perspective feels somewhat unbalanced.

We thank the Reviewer for a thoughtful perspective on the possible role of Ago in Cic regulation and the involvement of additional mechanisms. Based on this suggestion, we significantly toned down the language in the Conclusion about the role of Ago, and suggest that other ubiquitin ligases are likely to contribute to Cic degradation. In addition, we put more emphasis on other modes of Cic regulation via phosphorylation, such as DNA binding, nucleoplasmic shuttling, and repressor activity, and added the relevant references. As a result, we believe that our current version of the Conclusion presents a more balanced view of Cic regulation via multisite phosphorylation, while still emphasizing degradation, as our experiments were mostly focused on this form of regulation.

Minor comments

The Cic C1 domain is mentioned in the Conclusion and highlighted in Fig. S2, but without any explanation of what it is. I suggest writing "...the conserved HMG box and C1 DNA binding domains..." (p. 9) and including the corresponding reference (Fores et al., 2017).

We have made the suggested change, and this sentence now reads, "The homology between the human and fly Cic orthologs is low outside the conserved HMG box and C1 domains that mediate DNA binding...", with the appropriate references cited (p. 10). The C1 domain is now also mentioned in the Introduction (p. 3).

Second decision letter

MS ID#: jcs.264327R1

MS TITLE: ERK inhibits Capicua repressor function via multisite phosphorylation

AUTHORS: Sayantanee Paul; Khandan Ilkhani; Nathan Strozewski; Liu Yang; David W Denberg; Wootchelmine Christalin; Vanessa Locke; Stanislav Y Shvartsman; Alexey Veraksa

ARTICLE TYPE: Short Report

Dear Dr Veraksa,

We have now reached a decision on the above manuscript.

As you will see, the reviewers gave favourable reports. However, both referees each raised one critical point that will require minor amendments to your manuscript. I hope that you will be able to carry these out because I would like to be able to accept your paper.

Second revision

This is a second Response to Reviewers' comments for the manuscript Paul et al., jcs.264327, titled "ERK inhibits Capicua repressor function via multisite phosphorylation". Reviewers' comments are italicized.

Reviewer 1: I appreciate the authors' thorough and thoughtful responses to my previous comments. The revisions have significantly improved the clarity and overall quality of the manuscript. The additional explanations and changes have strengthened the manuscript and resolved the issues I previously raised.

*I have no further major concerns and believe the manuscript is close to being suitable for publication, pending one final modification. Specifically, clarification and revision of the quantification and statistical analysis presented for the western blot in Figure 4 are needed. The control group is shown without error bars and appears to be fixed at 1. Please clarify whether statistical analyses were performed on independent biological replicates prior to normalisation. If normalisation was performed per replicate, the control group should still display variance, and appropriate error bars (SD or SEM) should be shown. If the control values were fixed to 1 without variance (e.g., by dividing by an averaged control value), a Student's *t*-test would not be appropriate. In that case, the statistical analysis should be repeated using replicate-level data prior to normalisation.*

We agree with the Reviewer that we needed a different statistical approach to analyze these data. The experiment shown in Fig. 4A was done in 3 biological replicates performed on different days. We have now re-quantified all the data and analyzed the data using the one-sample *t*-test, which is the appropriate statistic to use in this case (for an example of a similar analysis, please see Bang et al., 2022, PMID 35869044).

Our null hypothesis was that there was no change in Cic level after exposure to MG132, meaning that the ratio of band intensities for +MG132 to -MG132 samples would be 1. We had to use this approach because our baselines were different for the three biological replicates, so we could not directly use the raw values for comparisons. Instead, we calculated these ratios within each western blot, and then used the ratios to compute one-sample *t*-tests separately for each Cic isoform against the baseline of 1. After calculating the *t*-statistics, *p*-values were derived using the T.DIST.RT function in Excel. Applying this approach to our results, for WT Cic such one-sample *t*-test had a $p < 0.05$ (mean ratio significantly different from 1), but for 6A or 20A it was > 0.05 (not significantly different from 1). The new data were replotted and are now shown in updated Fig. 4B, with a dotted line indicating the common baseline of 1 for all three sets of ratios. We have also updated the Fig. 4 figure legend and Materials and Methods with these details of statistical analysis.

Reviewer 2: The authors present an improved version of the manuscript, with new figures and a more balanced final discussion. The results are clear and interesting and, in my opinion, the work merits publication in Journal of Cell Science.

Nevertheless, I am not fully convinced by the statement on p. 9 that the mutated sites in Cic6A and Cic(7-9)A "appear to be dominant in mediating ERK-dependent downregulation." As written,

this implies that these sites account for most of the regulatory effect, with other phosphosites contributing smaller roles. However, because the different phenotypic effects are difficult to quantify and compare, I do not think the results unequivocally support this interpretation. It is possible, for example, that the defects observed in Figs. 1H, 2C, and 2O reflect a relatively modest gain of Cic function compared with the increase in activity that leads to the phenotypes shown in Figs. 2D, 2E, and 2Q. I feel that a wording such as "These regions in Cic play a substantial role in mediating ERK-dependent downregulation" would be more appropriate.

We agree with the Reviewer that the indicated statement should be toned down, and we have updated that sentence using the language proposed by the Reviewer.

Third decision letter

MS ID#: jcs.264327R2

MS Title: ERK inhibits Capicua repressor function via multisite phosphorylation

Authors: Sayantane Paul; Khandan Ilkhani; Nathan Strozewski; Liu Yang; David W Denberg; Wootchelmine Christalin; Vanessa Locke; Stanislav Y Shvartsman; Alexey Veraksa

Article Type: Short Report

Dear Dr Veraksa,

I am happy to tell you that your manuscript has been accepted for publication in Journal of Cell Science, pending standard publication integrity checks.